# Missense Variants of Uncertain Significance: A Powerful Genetic Tool for Function Discovery with Clinical Implications

**DOI:** 10.3390/cancers13153719

**Published:** 2021-07-23

**Authors:** Gaetana Sessa, Åsa Ehlén, Catharina von Nicolai, Aura Carreira

**Affiliations:** 1Institut Curie, Université PSL, CNRS UMR3348, INSERM U1278, 91400 Orsay, France; gaetana.sessa@curie.fr (G.S.); asa.ehlen@gmail.com (Å.E.); catharina@von-nicolai.de (C.v.N.); 2Université Paris-Saclay, CNRS UMR3348, INSERM U1278, 91400 Orsay, France

**Keywords:** variants of uncertain significance, breast cancer susceptibility, functional assays

## Abstract

**Simple Summary:**

Variants of uncertain significance in the breast cancer susceptibility gene *BRCA2* represent 50–80% of the results from genetic testing. These mutations may lead to the dysfunction of the gene, thus conferring breast cancer predisposition; however, because they are rare and their impact on the function is not easy to predict, their classification into benign or pathogenic variants remains a challenge. By focusing on three specific rare missense variants identified in breast cancer patients, in this review, we discuss how the functional evaluation of this type of variants can be used to reveal novel activities of BRCA2. Based on these findings, we suggest additional functional tests that might be required for accurate variant classification and how their characterization may be leveraged to find novel clinical strategies for patients bearing these mutations.

**Abstract:**

The breast cancer susceptibility gene *BRCA2* encodes a multifunctional protein required for the accurate repair of DNA double-strand breaks and replicative DNA lesions. In addition, BRCA2 exhibits emerging important roles in mitosis. As a result, mutations in *BRCA2* may affect chromosomal integrity in multiple ways. However, many of the *BRCA2* mutations found in breast cancer patients and their families are single amino acid substitutions, sometimes unique, and their relevance in cancer risk remains difficult to assess. In this review, we focus on three recent reports that investigated variants of uncertain significance (VUS) located in the N-terminal region of BRCA2. In this framework, we make the case for how the functional evaluation of VUS can be a powerful genetic tool not only for revealing novel aspects of BRCA2 function but also for re-evaluating cancer risk. We argue that other functions beyond homologous recombination deficiency or “BRCAness” may influence cancer risk. We hope our discussion will help the reader appreciate the potential of these functional studies in the prevention and diagnostics of inherited breast and ovarian cancer. Moreover, these novel aspects in BRCA2 function might help find new therapeutic strategies.

## 1. Introduction

Germline mono-allelic mutations in the tumor suppressor genes *BRCA1/2* (breast cancer susceptibility gene 1/2) predispose to breast and ovarian cancer [1], and in the case of *BRCA2*, also confer moderate risk to other cancers such as pancreatic and prostate cancer [2]. Bi-allelic mutations in *BRCA2/FANCD1* are at the origin of a subgroup of the rare genetic disease Fanconi anemia, which confers cancer susceptibility in children [3]. BRCA2 variants predisposed to cancer are inactivating. These, in most cases, introduce a stop codon either directly or indirectly through outframe insertions or deletions, or aberrant splicing. Based on a combination of different parameters such as clinical data, family history, and the effect on the gene’s function, a set of variants are classified as benign/likely benign, or pathogenic/likely pathogenic [4]. However, screening *BRCA1/2* also resulted in the discovery of thousands of generally rare missense substitutions, the influence of which on cancer risk is unknown. These so-called variants of uncertain significance (VUS) represent 50–80% of the results from genetic testing according to public databases (Table 1), which challenge the appropriate cancer risk management strategies for the individuals carrying these mutations [5,6]. Thus, the evaluation of VUS has important clinical implications.

BRCA2 exerts its tumor suppressor activity through its role in the maintenance of genomic stability, among which, DNA repair by homologous recombination (HR) [7] and the protection of stalled replication forks from aberrant nucleolytic degradation [8,9,10] are well characterized.

At 3418 amino acids, BRCA2 contains two well-defined domains, the BRC repeats implicated in binding the recombinase RAD51 and the only folded domain of BRCA2, the C-terminal DNA binding domain, both of which are important to exert its recombination activity [11,12]. In contrast, other parts of the protein remain poorly characterized, challenging the prediction of the impact of missense variants localized outside these two domains on BRCA2 function.

Tumor formation in *BRCA2* mutation carriers generally requires the inactivation of the wild-type allele, often detected by a loss of heterozygosity (LOH) [13]. Thus, the functional characterization of variants is generally assessed in homozygosis [14].

BRCA2 harbors several intrinsically disordered regions (IDRs) [15]; these IDRs serve as hubs for protein–protein and protein–DNA interactions and are subjected to cell-cycle-dependent post-translational modifications (PTMs) that regulate BRCA2’s activity. However, the function of many of these interactions and PTMs are ill-defined. Two recent studies have shed light on additional mechanisms by which BRCA2 ensures genomic integrity involving novel interactions and PTMs through the N-terminal IDR of the protein: the resolution of DNA-RNA hybrids at DNA double-strand breaks (DSBs) [16] and the promotion of proper chromosome alignment during cell division [17]. Common to the two studies is the VUS BRCA2-T207A (Table 2), an unclassified missense variant identified in breast cancer patients. Sessa and colleagues demonstrated that BRCA2 stimulates the helicase activity of the DEAD-box RNA helicase DDX5, enhancing the resolution of DNA-RNA hybrids at DSBs. BRCA2-T207A reduces the binding of DDX5 to the hybrids, leading to unresolved DNA-RNA hybrids and consequently less efficient repair of DSBs by HR. The mitotic phenotype of cells expressing the same variant was studied by Ehlén and colleagues, which led to the finding that BRCA2 is phosphorylated at this residue by PLK1 in mitosis, promoting chromosome alignment and faithful segregation; a mechanism that appears uncoupled to the interphase function of BRCA2 [17].

In an earlier study, Nicolai et al. used the breast cancer variant C315S (Table 2) to characterize a novel DNA binding site in the N-terminus of BRCA2 (NTD) [20]. Nicolai and colleagues showed that the C315S variant disrupts the association of the NTD with dsDNA that they showed was required for the stimulation of RAD51-mediated recombination. These three reports, among others, illustrate the power of missense variants as a tool to characterize protein activities. In turn, the study of separation of function variants of BRCA2, and of other tumor suppressors, identified in patients may provide insights on the specific function(s) that drive tumorigenesis when defective. We argue that these novel functions may expand the number of assays to be performed for VUS accurate reclassification. In this review, we describe the findings of these three reports in this context and discuss the implications for the classification of VUS, cancer risk assessment, and patient stratification management (Figure 1).

## 2. Functional Impact of BRCA2-T207A in Mitosis

BRCA2 has been implicated in mitosis, specifically through its localization to the midbody, where it serves as a scaffold for proteins that promote cytokinesis [21,22,23]. This function requires the phosphorylation of BRCA2 by cyclin-dependent kinase 1 (CDK1) and polo-like kinase 1 (PLK1) [22] and entails several interactions, including with Filamin A, which are compromised in several BRCA2 VUS. As a consequence, cells bearing these variants in homozygosis (ex. E3002K) exhibit binucleation and cytokinesis failure [21].

A recent study found that PLK1 binds and phosphorylates BRCA2-T207 in mitosis. The authors showed that human DLD1 (colorectal cancer) *BRCA2*-deficient cells stably transfected with the cDNA of the *BRCA2* breast cancer variants S206C or T207A (Table 2), which abolish PLK1 phosphorylation at the latter residue, share similar mitotic phenotypes such as unstable microtubule–kinetochore attachments that result in elevated levels of misaligned chromosomes. As a consequence, cells expressing S206C and T207A increased the number of lagging chromosomes and aneuploidy. Remarkably, these variants partially reduced the frequency of chromosome bridges observed in *BRCA2*-deficient cells [24,25], which arises as a consequence of accumulated replication or recombination intermediates, ultimately leading to failure to complete cytokinesis. The delayed mitosis, together with the defects in chromosome alignment and segregation without apparent impairment of the interphase functions of BRCA2, suggested that the dysfunction of these cells was not a consequence of endogenous replication stress or DNA repair defects, but rather of a specific function of BRCA2 in mitosis.

Taking advantage of the variants S206C and T207A, the authors revealed a novel docking site of PLK1 at T207, the phosphorylation of which facilitates the formation of a complex consisting of BRCA2, PLK1, BUBR1, and PP2A-B56 in mitotic cells. They found that within this tetrameric complex, BRCA2 acts as a molecular platform that facilitates the recruitment and phosphorylation of BUBR1 by PLK1 enhancing its interaction with the phosphatase PP2A-B56. This interaction is critical for the regulation of kinetochore-microtubule attachments and proper chromosome alignment (Figure 2A) [26]. Thus, the investigation of BRCA2 missense variants S206C and T207A uncovered a direct role of BRCA2 in the alignment of chromosomes and on faithful cell division.

The differences between the mitotic phenotype of cells expressing full-length BRCA2 with a single amino acid substitution, S206C or T207A, and cells deficient in BRCA2, clearly illustrate the importance of the study of missense mutations to understand protein function; in this case, by revealing a role in chromosome alignment that otherwise would have been masked by the massive unresolved chromosome bridges arising from replication stress observed in *BRCA2*-deficient cells [17,24,25]. Indeed, cells expressing VUS S206C and T207A are not sensitive to PARP (poly(ADP-ribose) polymerase inhibition (PARPi), a chemotherapy drug currently used for the treatment of *BRCA1/2* mutated tumors [27], do not show features of replication stress, and display only mild sensitivity to induced DNA damage, the latter probably explained by the contribution of BRCA2 in resolving DNA-RNA hybrids at DSBs, as discussed below.

We argue that this notion (i.e., the significance of characterizing missense variants for function discovery) might be particularly relevant in the case of large and multi-functional proteins such as BRCA1 or BRCA2.

## 3. Impact of BRCA2-T207A on DNA-RNA Hybrid Resolution at DSBs

DNA-RNA hybrids are stable structures that form in a variety of cellular processes such as transcription or replication. Increasing evidence indicates that DNA-RNA hybrids also form in the vicinity of DSBs present at transcribed regions and that these hybrids impact their repair. Recently, several studies have suggested that hybrids formed at DSBs via de novo transcription may act as intermediates and/or serve as regulators of the DNA repair pathway choice [28,29,30,31]. Other investigations have shown that DNA-RNA hybrids impede DSB repair by HR [16,32,33]. In both scenarios, however, DNA-RNA hybrids need to be removed to allow recombination-mediated DNA repair. A recent report demonstrated that BRCA2 physically binds to the RNA helicase DDX5 and showed that this interaction acts in the resolution of DNA-RNA hybrids associated with DSBs (Figure 2B) [16]. Sessa and colleagues’ results are consistent with DNA-RNA hybrids constituting an impediment for the repair of DSBs by HR and revealed that BRCA2 and DDX5 are active factors in their suppression. The authors observed that DDX5 associates with DNA-RNA hybrids forming at DSBs and BRCA2 can enhance this association. Mechanistically, BRCA2 was found to stimulate DDX5 unwinding activity on DNA-RNA hybrids by promoting its ATPase activity [16]. These findings shed light on the possible role of BRCA2 on DNA-RNA structures suggested by previous studies, which reported the accumulation of DNA-RNA hybrids in *BRCA2*-deficient cells [34].

Sessa and colleagues mapped BRCA2-DDX5 interaction and the helicase stimulatory activity to the first 250 aa of BRCA2. However, to directly determine the relevance of BRCA2-DDX5 interaction for the function of both proteins at DSB-associated DNA-RNA hybrids, identifying a mutation that impeded the interaction while minimizing the effect on other aspects of BRCA2 or DDX5 functions was critical. Scanning the BRCA2 sequence for VUS that increased the levels of DNA-RNA hybrids, they selected T207A, one of the two variants that they had previously found to alter chromosome alignment [17], as a candidate variant. The authors reported that in BRCA2-deficient cells stably expressing BRCA2-T207A, DDX5-BRCA2 interaction was indeed reduced and correlated with overall increased levels of DNA-RNA hybrids. Importantly, they showed that DNA-RNA hybrids were especially increased in cells expressing BRCA2-T207A subjected to induced DSBs, suggesting that DSB-associated DNA-RNA hybrids were the target of BRCA2-DDX5 interaction. Consistent with this, BRCA2-T207A cells exhibited a lower occupancy of DDX5 at DNA-RNA hybrid-associated DSBs and at laser-induced DNA damage, corroborating that BRCA2-DDX5 interaction promotes the localization/retention of DDX5 at DNA-RNA hybrids accumulated at DNA breaks. These results prompted the authors to test the DSB repair efficiency in these cells. Interestingly, cells expressing BRCA2-T207A reduced the kinetics of appearance of RAD51 repair foci, a phenotype that was also observed upon DDX5 depletion, suggesting that the timely repair of at least a subset of DSBs by HR requires BRCA2-DDX5 interaction. Importantly, the fact that the delayed kinetics observed in cells expressing BRCA2-T207A was partially rescued by RNAseH overexpression (an enzyme that degrades the RNA moiety of DNA-RNA hybrids) indicated that the DNA-RNA hybrid resolution activity of BRCA2-DDX5 directly impacts HR repair. Indeed, the DDX5 unwinding activity, which was stimulated by BRCA2 in vitro, could be essential for the resolution of DNA-RNA hybrids formed at DSBs and for efficient repair by HR. The fact that a recombinant fragment of BRCA2 comprising the T207A variant precluded DDX5 unwinding activity on synthetic DNA-RNA hybrids in vitro strongly supports this idea.

These findings may have several clinical implications: i. BRCA2-T207A, without altering the canonical BRCA2 domains directly implicated in the HR function (the BRC repeats and the CTD), imposes deleterious effects on RAD51-mediated DSB repair. To our knowledge, this is the first time this kind of indirect effect has been observed for BRCA2 missense VUS. Interestingly, the influence on HR was not detected by the classical GFP-HR reporter assay [17] currently used in the functional evaluation of VUS suggesting that complementary assays such as the kinetics of appearance of RAD51 repair foci might be needed for the accurate assessment of HR proficiency [14]. ii. High levels of R-loops arising from endogenous or environmental aldehydes have been proposed to be a cause of tumorigenesis in *BRCA2* mutation carriers [35]. Together, these reports suggest that testing the levels of DNA-RNA hybrids could be a complementary method to reveal the functional impact of missense variants and evaluate their pathogenicity. ii. BRCA2-T207A also affects chromosome alignment and segregation, leading to aneuploidy, a feature observed in BRCA2 breast tumors, a compound phenotype that underscores the possible clinical relevance of this variant.

## 4. The C315S VUS Cannot Stimulate the Recombination Activity of RAD51

During DSB repair by HR, BRCA2 binds at the ssDNA/dsDNA junction generated after processing of the break by nucleases and facilitates the assembly of a stable RAD51 nucleo-protein filament on ssDNA (Figure 2C). This event requires the displacement of the replication protein A (RPA) that coats the ssDNA immediately after resection, which is facilitated by BRCA2. A stable RAD51 filament is the functional unit and rate-limiting step necessary to invade the template strand and repair the break [36,37,38]. The requirement of the C-terminal DNA binding domain (CTD) for recombination was challenged by a study reporting the identification of *BRCA2*-deficient clones with acquired resistance to PARPi. Interestingly, some of these clones lacked the entire CTD and yet they were HR proficient [39], as measured by the classical DR-GFP reporter assay [40]. These results and the study of truncating versions of BRCA2 lacking the CTD that maintain their HR capacity [41] suggested that another part of the protein could be at play to restore HR function and underlie the PARPi resistance observed in those clones.

Previous reports from the Holloman group had identified a second DNA binding domain in the region between the unique BRC repeat and the CTD of the ortholog of BRCA2 in the fungus *Ustilago maydis*, Brh2, providing an important clue [42]. In 2016, Nicolai and colleagues described an additional DNA binding domain in the N-terminus of human BRCA2 upstream of the BRC repeats [20]. By expressing different fragments of the N-terminal region of the protein, Nicolai et al. demonstrated that the fragment containing aa 250–500 of BRCA2 (BRCA2-NTD) was sufficient to bind DNA in vitro.

Surprisingly, electrophoretic mobility shift assays (EMSAs) performed with the recombinant NTD and the canonical CTD fragments in isolation on different DNA substrates revealed that the NTD bound with higher affinity to all the synthetic DNA substrates tested compared to the CTD fragment. Remarkably, unlike the CTD, the NTD bound readily to dsDNA, revealing an important clue to their possible specificities. To characterize the function of this newly identified DNA binding domain, the authors tested the NTD capacity to stimulate the recombination activity of RAD51. As previous reports had described that BRCA2 required a minimum of one BRC repeat and a DNA binding domain to exert its mediator function in HR [43,44], the authors tested a fusion protein containing the NTD and BRC4. Indeed, the NTD-BRC4 promoted RAD51-mediated recombination in vitro, and, more strikingly, the NTD per se was sufficient to stimulate the strand exchange activity of RAD51 overcoming the inhibitory action of RPA. To obtain mechanistic insights on this function, the authors used several VUS located in the NTD region. BRCA2-C315S, currently classified as benign/likely benign in the ClinVar database as of 2020 (Table 2), turned out to be particularly revealing. Indeed, the C315S missense mutation specifically reduced the dsDNA binding activity of the NTD without altering its ssDNA binding affinity. This separation of function variant allowed the authors to demonstrate that the dsDNA binding activity of the NTD was required to stimulate RAD51 strand exchange activity at ss/dsDNA containing substrates such as those expected upon resection at DSB or at ssDNA gaps present at stalled replication forks. Whether this missense variant alters the recombination function of BRCA2 in the context of the full-length protein and cells or whether its recombination activity observed in vitro is particularly relevant for specific DNA damage conditions in vivo needs further investigation.

## 5. Conclusions

Overall, these reports underscore the difficulty of the interpretation of missense variants of unknown clinical significance and support the notion that not only a combination of tools (genetic, in silico prediction, co-segregation analysis) but also a combination of functional assays in addition to the standard HR assay might be required to properly assess the cancer risk of an individual carrying these mutations.

The accurate classification of rare missense substitution variants is further challenged by the finding that some of them may show a compound effect on function (such as T207A in BRCA2) [16,17] or a compound effect on function and splicing (such as S206C in BRCA2) (Table 2) [17,18]. Furthermore, some missense variants may exhibit partially reduced activity sufficient to confer a moderately increased risk of breast cancer, as shown for R1699N substitution in BRCA1 or Y3035S in BRCA2 [45]. These and other reports characterizing BRCA1/2 missense variants provide examples of how the study of these missense substitutions, although rare, are fundamental to dissect the activities of complex tumor suppressor proteins such as BRCA1/2, and in some cases, they may challenge the current classification of certain missense variants. For example, the prominent activity of BRCA2 in mitosis and the phenotype conferred by several VUS as mentioned above might be a function to consider when evaluating *BRCA2* VUS. This framework is equally relevant for BRCA1/2 bi-allelic mutations, such as those observed in Fanconi anemia patients [46].

The advent of base-editing technologies applied to the functional interrogation of variants will undoubtedly accelerate the impact of these studies on human health. Notably, saturation genome editing was used to assess the effect on function of 4000 single nucleotide variants (SNVs) in BRCA1 [47]. More recently, base-editing has been used to determine the impact on function of SNV of DNA repair genes such as BRCA1/2, and for the identification of VUSs with pathogenic potential [48].

We believe that these studies, in addition to the strong international efforts on variant classification [49], will be determinant for the accurate assessment of the cancer risk of individuals carrying these mutations, clinical decision making for cancer patients, and will likely reveal new cancer vulnerabilities for improved treatment response (Figure 1). For example, some of the variants mentioned in this review altering mitotic functions such as chromosome alignment or cytokinesis are not predicted to confer sensitivity to DNA damage or PARPi; thus, alternative strategies leveraging on their mitotic defects may increase treatment efficacy.

Given the toxicity and/or resistance to chemotherapy treatment, such as those observed with PARPi, it would be interesting to target BRCA2 interactions to use in combination with radio- or chemotherapy. In this regard, recent work has provided evidence on the synergy between BRCA2-RAD51 interaction inhibitors with PARPi [50,51].

Finally, these studies in combination with in vivo model systems should shed light on the fundamental question: which of these functions, when defective, drive tumorigenesis.

## Figures and Tables

**Figure 1 cancers-13-03719-f001:**
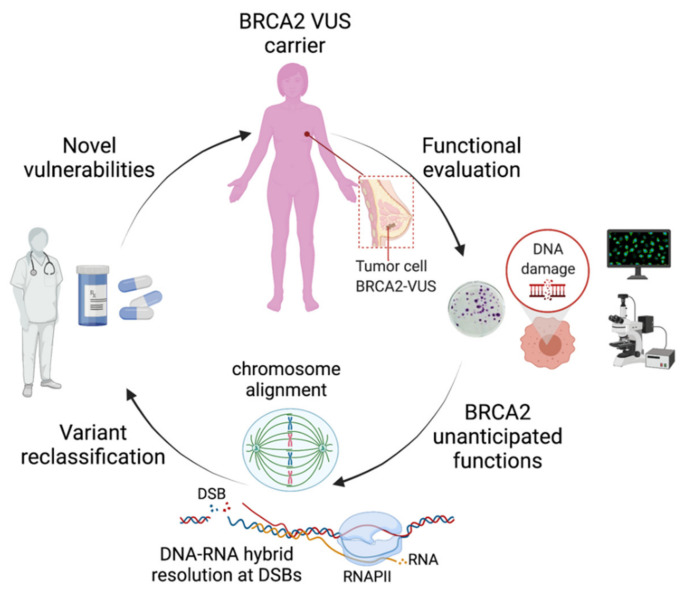
Schematic representation of the impact of the functional characterization of VUS identified in *BRCA2*-breast cancer patients on the discovery of unanticipated functions. These novel functions may have important clinical implications on the reclassification of VUS, as they may provide elements to revisit the functional assays best adapted for their evaluation. For the patients, the functional evaluation will improve the accurate classification of VUS for cancer risk assessment and may lead to the discovery of novel vulnerabilities that could be exploited for precision medicine. DSBs: DNA double-strand breaks; RNAPII: RNA polymerase II. Figure created with BioRender.com (accessed on 1 April 2021).

**Figure 2 cancers-13-03719-f002:**
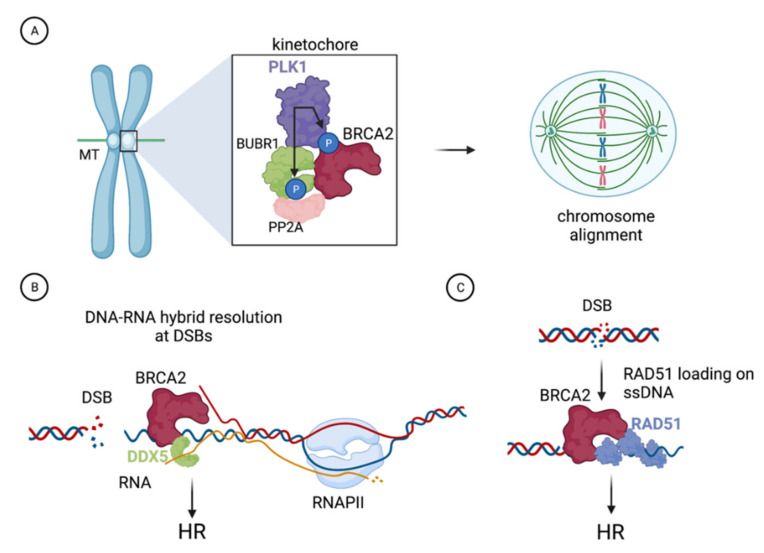
Example of functions of BRCA2 characterized with the help of the VUS as described in the text. (**A**). BRCA2 is phosphorylated by PLK1 at the kinetochore where it forms a complex with PLK1, BUBR1, and PP2A. This complex, which is impaired in cells expressing BRCA2-S206C or BRCA2-T207A VUS that cannot be phosphorylated by PLK1 at T207, is required for the stability of kinetochore microtubules (MTs) attachments and proper chromosome alignment. (**B**). BRCA2 stimulates the DNA-RNA hybrid unwinding efficiency of the RNA helicase DDX5 by enhancing its ATPase activity; this interaction facilitates DNA-RNA hybrid resolution at DNA double-strand breaks (DSBs) and is defective in the variant BRCA2-T207A. (**C**). BRCA2 comprises a second DNA binding domain in its N-terminal region (NTD). The NTD can stimulate the homologous recombination (HR) activity of RAD51 in vitro, a function that is reduced in the variant BRCA2-C315S, which is defective in dsDNA binding. Figure created with BioRender.com (accessed on 1 April 2021).

**Table 1 cancers-13-03719-t001:** The number of different *BRCA1/BRCA2* variants reported in public databases.

Database	Benign/Likely Benign	VUS	Pathogenic	Total # of Different Variants
BRCAexchange ^1^	1430 (6.3%)	32,978 (81.6%)	2672 (12.1%)	40,416
ClinVar ^2^	5046 (20.2%)	12,648 (50.7%)	7264 (29.1%)	24,964

^1^https://brcaexchange.org/ (accessed on 1 April 2021). ^2^
https://www.ncbi.nlm.nih.gov/clinvar/ (accessed on 1 April 2021); #: number.

**Table 2 cancers-13-03719-t002:** Missense VUS reported altering the three BRCA2 residues discussed in this review. VUS protein and DNA nomenclature, number of records in the ClinVar database, current classification, and affected function are indicated together with the original link to the ClinVar database.

VUS	ClinVar Records	Clinical Classification (*)	Phenotype and Function Affected by VUS/NCBI Link
S206C (c.617C > G)	1	N/A	Misaligned chromosomes, segregation errors, aneuploidy [17]. Aberrant splicing [18] https://www.ncbi.nlm.nih.gov/clinvar/variation/52026/ (accessed on 1 April 2021)
S206Y (c.617C > A)	2	Uncertain significance (2)	N/A https://www.ncbi.nlm.nih.gov/clinvar/variation/485433/ (accessed on 1 April 2021)
T207A (c.619A > G)	4	Uncertain significance (4)	Misaligned chromosomes, segregation errors, aneuploidy [17]. Unresolved DNA-RNA hybrids, delayed DSB repair https://www.ncbi.nlm.nih.gov/clinvar/variation/52028/ (accessed on 1 April 2021)
T207I (c.620C > T)	7	Uncertain significance (7)	Aberrant splicing [19] https://www.ncbi.nlm.nih.gov/clinvar/variation/186155/ (accessed on 1 April 2021)
C315S (c.943T > A)	14	Benign (3) Benign/Likely benign (5) Likely benign (6)	Reduced in vitro RAD51-mediated recombination activity at ss/dsDNA containing DNA substrates https://www.ncbi.nlm.nih.gov/clinvar/variation/38241/ (accessed on 1 April 2021)
C315Y (c.944G > A)	1	Uncertain significance (1)	N/A https://www.ncbi.nlm.nih.gov/clinvar/variation/1044855/ (accessed on 1 April 2021)

* number of submissions; N/A: not available.

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
