# Peer review of "Missense Variants of Uncertain Significance: A Powerful Genetic Tool for Function Discovery with Clinical Implications"

_cancers, 2021, doi:10.3390/cancers13153719_

Round 1

Reviewer 1 Report

Sessa G et al. introduced novel roles of missense variants of uncertain significance (VUSs) of BRCA2 in mitosis and double strand break (DSB) repair.  The contents of this review are well described incorporating their own data and other researchers’ work. Regrettably, they did not show a figure to visually explain the novel roles of VUSs of BRCA2 in mitosis and DSB repair.  Figure 1 is excellent but is somewhat too general, so in addition to Figure 1, they should add another figure, in which the roles of BRCA2 VUSs on faithful mitosis and on DSB repair should be illustrated more specifically at the molecular level with other important molecules described in the text, e.g., PLK1, DDX5, and RAD51.

The authors describe that the VUS BRCA-T207A functions on misaligned chromosome, segregation errors and aneuploidy formation on the one hand, and on delayed DSB repair via unresolved DNA-RNA hybrids on the other.  Please mention if these molecular events in two dimensions, i.e., mitosis and DSB repair, are closely related or independent. 

Author Response

Reviewer #1

Comments and Suggestions for Authors

Sessa G et al. introduced novel roles of missense variants of uncertain significance (VUSs) of BRCA2 in mitosis and double strand break (DSB) repair.  The contents of this review are well described incorporating their own data and other researchers’ work. Regrettably, they did not show a figure to visually explain the novel roles of VUSs of BRCA2 in mitosis and DSB repair.  Figure 1 is excellent but is somewhat too general, so in addition to Figure 1, they should add another figure, in which the roles of BRCA2 VUSs on faithful mitosis and on DSB repair should be illustrated more specifically at the molecular level with other important molecules described in the text, e.g., PLK1, DDX5, and RAD51.

Thanks for the suggestion. We have now added a new figure (Fig. 2) showing the specific function of BRCA2 in mitosis, DNA-RNA hybrid resolution and the stimulatory role during homologous recombination that have been characterized using BRCA2 breast cancer variants.

The authors describe that the VUS BRCA-T207A functions on misaligned chromosome, segregation errors and aneuploidy formation on the one hand, and on delayed DSB repair via unresolved DNA-RNA hybrids on the other.  Please mention if these molecular events in two dimensions, i.e., mitosis and DSB repair, are closely related or independent. 

As it is mentioned in the text, lanes 78-79 and 122-125, the mitotic function of BRCA2 and its role at DSBs or replication in interphase appear to be independent the former involving its phosphorylation by PLK1.

Reviewer 2 Report

Sessa et al., reviewed multiple BRCA2 mutations that are found as variants of uncertain significance (VUS) and the mechanistic characterizations together with those significance. Basically, the manuscript is conceptually quite interesting and important. In fact, the characterization of VUS is critical for the therapeutic strategy in each cancer. However, those mutations reviewed in this manuscript are very limited and are likely done by the authors’ own previous studies, in most. I feel this is a problem as this type of review manuscript, because this is not the short commentary. Overall, I do not think this manuscript is publishable in the current format; therefore, I would strongly recommend to revise.

Major issue;

#1: The major problem of this manuscript is that BRCA2 VUS mutations specifically reviewed in this manuscript are very limited and biasedly selected. One thing what I would recommend is to draw the BRCA2 mutations on the BRCA2 domain map with summarizing those mutations reported in the public database together with VUS information. That way, we can see where VUS are located, and which types of mutations the authors are specifically reviewing. In fact, since public database analyses are somehow already included in Table 1, this must be quite straight forward.

#2: One of the important questions in the field is whether those cancers with BRCA2 VUS mutations are sensitive to PARP inhibitors, or not. I would strongly recommend to add this information, such as in Table 2. That will be helpful.

Mini issue;

Line 15: VUS must be defined.

Table 1: 6,3 %, 20,2 %, 81,6, % and 50,7 % should be 6.3 %, 20.2 %, 81.6, % and 50.7 %.

Line 71: Ehlen et al., this should be referenced.

Line 161: This sentence also should be referenced.

Author Response

Reviewer #2

Comments and Suggestions for Authors

Sessa et al., reviewed multiple BRCA2 mutations that are found as variants of uncertain significance (VUS) and the mechanistic characterizations together with those significance. Basically, the manuscript is conceptually quite interesting and important. In fact, the characterization of VUS is critical for the therapeutic strategy in each cancer. However, those mutations reviewed in this manuscript are very limited and are likely done by the authors’ own previous studies, in most. I feel this is a problem as this type of review manuscript, because this is not the short commentary. Overall, I do not think this manuscript is publishable in the current format; therefore, I would strongly recommend to revise.

Major issue;

#1: The major problem of this manuscript is that BRCA2 VUS mutations specifically reviewed in this manuscript are very limited and biasedly selected.

The number of BRCA2 VUS that have been functionally characterized in the literature is very extensive and this review is not intended to compile this information. Instead, we thought it would be useful to highlight some examples that we and others have described that have revealed novel aspects of BRCA2 function that might be relevant in the clinic.

One thing what I would recommend is to draw the BRCA2 mutations on the BRCA2 domain map with summarizing those mutations reported in the public database together with VUS information. That way, we can see where VUS are located, and which types of mutations the authors are specifically reviewing. In fact, since public database analyses are somehow already included in Table 1, this must be quite straight forward.

As mentioned above, we have mainly focused on three variants located in two sites of the N-terminal region, since the other variants affect the same amino acids we deemed that a table (Table 2) would be more informative than a scheme of the protein.

#2: One of the important questions in the field is whether those cancers with BRCA2 VUS mutations are sensitive to PARP inhibitors, or not. I would strongly recommend to add this information, such as in Table 2. That will be helpful.

Thanks for the suggestion. This information is known for S206C and T207A variants as we have published (Ehlen et al 2020) so we have now included this information in Page 5, new lane 188. In the case of C315S, this is currently under investigation.

Mini issue;

Line 15: VUS must be defined. DONE

Table 1: 6,3 %, 20,2 %, 81,6, % and 50,7 % should be 6.3 %, 20.2 %, 81.6, % and 50.7 %. DONE, thanks

Line 71: Ehlen et al., this should be referenced. Done, thanks.

Line 161: This sentence also should be referenced. We have added this reference (now lane 213)